# Enrichment of *Cis*-Acting Regulatory Elements in Differentially Methylated Regions Following Lipopolysaccharide Treatment of Bovine Endometrial Epithelial Cells

**DOI:** 10.3390/ijms25189832

**Published:** 2024-09-11

**Authors:** Naveed Jhamat, Yongzhi Guo, Jilong Han, Patrice Humblot, Erik Bongcam-Rudloff, Göran Andersson, Adnan Niazi

**Affiliations:** 1Department of Animal Biosciences, Swedish University of Agricultural Sciences, P.O. Box 7023, SE-75007 Uppsala, Sweden; 2Department of Clinical Sciences, Swedish University of Agricultural Sciences, P.O. Box 7023, SE-75007 Uppsala, Sweden; 3SLU-Global Bioinformatics Centre, Swedish University of Agricultural Sciences, P.O. Box 7023, SE-75007 Uppsala, Sweden

**Keywords:** transcription factor binding sites, *cis*-acting regulatory element, differential DNA methylation, gene regulation, transcriptional networks, LPS

## Abstract

Endometritis is an inflammatory disease that negatively influences fertility and is common in milk-producing cows. An in vitro model for bovine endometrial inflammation was used to identify enrichment of *cis*-acting regulatory elements in differentially methylated regions (DMRs) in the genome of in vitro-cultured primary bovine endometrial epithelial cells (bEECs) before and after treatment with lipopolysaccharide (LPS) from *E. coli*, a key player in the development of endometritis. The enriched regulatory elements contain binding sites for transcription factors with established roles in inflammation and hypoxia including NFKB and Hif-1α. We further showed co-localization of certain enriched *cis*-acting regulatory motifs including ARNT, Hif-1α, and NRF1. Our results show an intriguing interplay between increased mRNA levels in LPS-treated bEECs of the mRNAs encoding the key transcription factors such as AHR, EGR2, and STAT1, whose binding sites were enriched in the DMRs. Our results demonstrate an extraordinary *cis*-regulatory complexity in these DMRs having binding sites for both inflammatory and hypoxia-dependent transcription factors. Obtained data using this in vitro model for bacterial-induced endometrial inflammation have provided valuable information regarding key transcription factors relevant for clinical endometritis in both cattle and humans.

## 1. Introduction

Correctly regulated transcription is crucial in multicellular organisms with different cell types. The characterization of the *cis*-acting DNA sequences and *trans*-acting factors that are required to control gene regulatory networks is, therefore, a highly prioritized research area in genomics, transcriptomics, and epigenomics, in all transcriptional networks including those that control inflammatory responses [1]. Regulation of RNA polymerase II-dependent transcription of protein-coding genes is fundamental for allowing control of both spatial and temporal gene expression and genetic variation in transcriptional regulatory regions in the genome is consequently important for the phenotypic diversity within populations.

A fundamental aspect of transcriptional regulation is the function of DNA-binding transcription factors during activation and repression of transcription. Another key aspect of transcriptional regulation is chromatin structure as a result of differential DNA methylation and histone modification [2,3]. At least 2000 known transcriptional regulatory proteins have been identified in the human proteome, including approximately 1600 DNA-binding transcription factors [4]. The processing of their transcripts often generates multiple alternatively spliced transcripts and intriguingly transcription factors themselves appear to be critically involved in regulating alternative splicing [5]. The complexity of different transcription factors is thus tremendous and also involves sophisticated post-translational modifications, such as glycosylation, phosphorylation, acetylation, citrullination, and ubiquitination, and more modifications that remain to be further defined. It is, however, clear that complex organisms have evolved a substantial variation in proteins controlling inducible, developmental-, cell type-, and tissue-specific gene expression. DNA-binding transcription factors are pivotal regulatory proteins that are required for control of transcriptional initiation, activation, and repression. They bind to specific DNA sequences to control the level of gene expression in a context-dependent and tissue-specific manner [6,7]. The transcriptional regulatory regions of protein-coding genes are classified as position-dependent promoters, and position-independent enhancers, silencers, insulators, and locus control regions (LCRs). All these regulatory regions contain recognition sites for transcription factors [8]. Transcription factors bind to these regulatory *cis*-acting DNA sequences referred to as Transcription Factor Binding Sites (TFBSs), which are short and often well-conserved sequence motifs [9]. Importantly, several such motifs are often clustered together and overlapping each other, forming a regulatory module [10,11]. This organization of regulatory regions allows for cooperative interactions between different transcription factors and a dynamic regulation depending of which group of transcription factors that occupy a given module [12]. Transcription factors are also by themselves modular proteins and most of them consist of a DNA-binding domain (DBD), a dimerization domain, and an activation or repression domain [6,13,14]. The DBD recognizes specific DNA sequence-binding motifs, which are often short palindromic sequences [15]. Around 25% of promoters in mammalian genomes has been estimated to contain regulatory motifs that are derived from transposable elements [4,16,17,18]. Close to 300,000 regulatory elements originate from insertions of mobile transposable DNA [19].

Transcription factors are classified into families on the basis of sharing similar types of DBD and/or dimerization domains. During evolution, these families were expanded and diverged, which caused organismal complexity that depends on the specific expression of each gene that is regulated by transcription factors [20]. The basic helix loop helix (bHLH) family comprises of more than 40 genes in *C. elegans* [21]. In mammalian genomes the number genes that encode bHLH transcription factors exceeds one hundred [22]. The exact time when a gene is transcribed and in what cell type and at what level of mRNA expression in multi-cellular organisms is controlled through the compound interaction of transcription factors and TFBSs [23]. For example, the role of AP-2 transcription factor has been reported in embryonic development by influencing the differentiation and proliferation of various different cell types [24]. The NF-kappaB (NFKB) and Estrogen receptor (ER) families are two other types of transcription factors of critical importance that regulate many genes involved in development, maintenance, immune response, and reproduction [25].

The prediction of potential TFBSs or regulatory regions using bioinformatics approaches at the genome-wide level is a prioritized area in genomics and epigenomics. Some of these TFBSs are often highly conserved in related species whereas others are more divergent [26]. The development of high-throughput Next Generation Sequencing (NGS) technologies has enabled whole genome DNA sequencing at high coverage at a low cost and in a short time [27,28]. The availability of whole genome DNA sequence datasets from a large number of different species has allowed the identification of potential transcriptional regulatory regions [29,30]. The available bioinformatics resources such as TRANSFAC [31], JASPAR [32], and ENCODE, Mouse ENCODE [33], Factorbook [4], and modENCODE (Model Organism Encyclopedia of DNA Elements) enable the assessment of functional elements in the genomes of both human and model organisms. However, our current knowledge regarding the dynamics and complexity of transcription factor binding to such motifs and the cooperative interactions at regulatory regions in most mammalian species such as the bovine genome is quite limited. Recently, a substantial step towards an in depth understanding of this was achieved by the Zoonomia Consortium [17]. Here the authors mined *cis*-regulatory sequences in genomes from more than 240 mammals.

Our group has intensively studied the impact of inflammation on the bovine endometrium while performing a large number of molecular studies, from both in vivo and in vitro models and the present bioinformatics study of transcription factor binding sites located in differentially methylated regions is a continuation of these to better understand the molecular nature of the early inflammatory response in the endometrium.

A healthy uterine endometrium with normal transcriptional regulation and gene expression is critical for successful implantation of the embryo and its growth and development to term. Bacterial infections of the uterus, often by *Escherichia coli* (*E. coli*), cause metritis and subclinical endometritis, which is a highly prevalent inflammatory disease of the endometrium that negatively influence implantation and reproductive performance of postpartum dairy cows. During the early stages of inflammation, the infected tissue responds by a massive activation of transcription and chromatin remodeling and gene expression of immune genes like pro-inflammatory cytokines and tissue remodeling genes [34]. The bovine endometrium is a complex and dynamic tissue with caruncular and inter-caruncular parts covered by a luminal epithelium. The inter-caruncular part consists mainly of glandular epithelial cells whereas the carunculum consists of stromal cells. In addition, this dynamic tissue may contain many other cell types, and the cellular content is heavily influenced by the estrous cycle, pregnancy, and inflammation. As described above, when the endometrium is challenged by bacterial infections leading to endometritis, the uterine receptivity is altered and initiation of pregnancy is compromised by the presence of high numbers of infiltrated immune cells [35,36].

Subclinical endometritis is a major problem in milk-producing cows. To study the effects of the initial phase of the inflammation of the endometrial epithelium, our group has established an in vitro model of bovine endometrial epithelial cells [36]. To enable this model, purified endometrial epithelial cells were cultured and treated with lipopolysaccharide (LPS) to mimic a bacterial infection as described [34]. The glandular epithelial cells and luminal epithelial cells are the two different epithelial cell types in the endometrium. In the bovine endometrium, glandular epithelial cells represent between 80% and 90% of the epithelial cell content whereas the content of luminal epithelial cells is between 10% and 20% [37]. Recently we performed transcriptomic and epigenomic analyses of purified bovine endometrial epithelial cells (bEECs) and identified differentially expressed genes (DEGs) and differentially methylated regions (DMRs) following *E. coli* LPS treatment of bEECs [34,38]. Furthermore, in the in vivo situation, transcriptomic profiling of mRNAs prepared from stromal, glandular, and luminal epithelial endometrial cells isolated by laser capture microdissection of endometrial tissue taken from cows with subclinical endometritis revealed distinct gene expression profiles in these cell types [37].

The aim of the current study was to identify and classify the enrichment of TFBSs present in the DMRs detected in these bEECs in response to *E. coli* LPS to obtain information about modular and cooperative nature of *cis*-regulatory elements responding to inflammation in endometrial epithelial cells at the genome-wide level. The basis for this approach is the 1291 DMRs that were identified in our earlier Reduced Representation Bisulfite Sequencing (RRBS) study [38]. Furthermore, the characterization of TFBSs in these LPS-induced DMRs will increase our understanding regarding the regulatory regions controlling differential gene expressing networks that we have identified by RNA-sequencing of the same LPS-treated cells [34]. Thus, the transcriptomics, epigenomics, and bioinformatics data obtained from these three in vitro studies will allow us to draw conclusions regarding the complexity of the regulatory transcriptional machinery activated during early stage of inflammation in the bEECs. Increased understanding of how transcription of genes involved in embryo-maternal interactions and how this genetic network is regulated during inflammation and negatively impact successful fertility may allow development of treatment regimen and increased reproductive success.

To further our understanding of the transcriptional regulatory networks activated during inflammatory responses we performed bioinformatics analyses to define the potential enrichment of transcription factor binding sites that were detected in the DMRs obtained following LPS-treatment of bEECs. Here we describe the nature of the transcription factor binding sites in regulatory regions controlling transcription during endometrial inflammatory responses using this in vitro model system.

## 2. Results

Previously, we used an in vitro model for bovine endometritis and identified 1291 differentially methylated regions (DMRs) in bovine endometrial epithelial cells treated with LPS using Reduced Representation Bisulfite Sequencing (RRBS) data [38]. To assess whether these DMRs have transcriptional regulatory potential, consensus transcription factor binding motifs within hyper- and hypo-methylated DMRs were searched for in databases with transcription factor binding profiles (see Section 4). The most significant consensus motifs located within both hyper-methylated and hypo-methylated DMRs are shown in Figure 1. We compared these motifs against databases of known transcription factor motifs to find regulatory sites. This analysis revealed binding sites for immunologically important transcription factors such as IRF1, IRF3, STAT1, and SOX15 [39,40,41].

To further evaluate the potential effects of altered DNA methylation on gene regulation, we examined both significantly hyper-methylated and hypo-methylated DMRs for known transcription factor binding motif enrichment using oPOSSUM [42]. In total, 73 distinct, significantly over-represented motifs were identified in the DMRs. Both hyper and hypo DMRs represented a different set of transcription factors with a minor overlap (Figure 2A). Hyper DMRs were enriched for 35 transcription factor motifs, whereas hypo DMRs were enriched for 45 motifs (Appendix A). Interestingly, both hyper and hypo DMRs were enriched for the binding sites of transcription factors involved in the maintenance of cell proliferation, cell migration, cell adhesion, inflammatory response, and reproductive functions. Most noteworthy among these were the five members of STAT family including STAT1, STAT2, STAT3, STAT5a/b, and STAT6. Other significantly over-represented transcription factor motifs include those that bind transcription factors that belong to the interferon regulatory factor family (e.g., IRF1, IRF2, and IRF7). Furthermore, binding sites for the early growth response family (EGR1, EGR2, EGR3), activating enhancer-binding protein 2 (TFAP2C), and aryl hydrocarbon receptor nuclear translocator (ARNT) were also shown to be enriched in the DMRs. Methylation of CpG sites (CpGs) affects both the chromatin structure and binding affinity of transcription factors—in some cases negatively and in others positively [43]. Therefore, we examined consensus sequences of the enriched binding sites in the DMRs to identify transcription factors that might directly be influenced by the methylation status at CpG dinucleotides. We found two distinct set of motifs harboring CpGs for both hyper and hypo DMRs (Figure 2B). Interestingly, motifs for the homeo-domain (HOX) protein family were found to contain demethylated CpGs. HOX proteins prefer to bind to methylated CpGs and play a pivotal role in embryonic and organismal development [43].

Next, we classified over-represented motifs into different classes of transcription factors. Transcription factors belonging to three different families, i.e., the basic helix-loop-helix family (BHLH), members of the homeo-domain family (HOX), and the basic leucine zipper family (bZIP), were largely over-represented (Table 1). Enrichment of TFBSs in the promoter regions defined as 1 kb upstream of the transcription start sites of differentially expressed genes (DEGs) reported in our previous work [34] were analyzed. In total, 182 motifs were significantly enriched, with many enriched in the promoter regions of both over- and under-expressed genes (Appendix A). There was an overlap of 27 TFBSs enriched both in DMRs and in the promoters of DEGs (Figure 2C). This supports the power of our in vitro model for bovine endometritis that has allowed the identification of transcription factors binding to these enriched motifs, a finding that could be relevant for the understanding of functional changes that develop during the initiation of the inflammatory response.

### 2.1. Co-Localization of Transcription Factor Cis-Regulatory Motifs

The motifs for enriched transcription factors were further scrutinized for flanking transcription factor binding sites (TFBSs) in the DMRs. Five significantly enriched transcription factors were chosen with hits in at least 20% of the total DMRs. Motifs for PAX2, HLTF, and MEIS1 were shared in DMRs that were hyper-methylated and motifs for AHR::ARNT, TCFL5, and HES1 were shared among the DMRs that were hypo-methylated (Figure 2D), whereas motifs for ARNT::HIF-1α, and NRF1 were found in both hypo- and hyper-methylated DMRs. Next, we investigated DMR sequences with significantly enriched motifs in both hyper and hypo DMRs for co-occurring binding sites for different transcription factors. We observed similar colocalization patterns of transcription factor motifs in both hyper and hypo DMRs with minor differences in regions enriched for ARNT::HIF-1α, NRF1, and FOXB1 (Figure 2E).

### 2.2. Effects of Methylation Changes on the Activity of Transcription Factors

To understand how methylation changes influenced interactions with the transcription factors activity, we used gene expression data from our previous work [34]. Among the transcription factor motifs enriched in the DMRs, a total of 37 genes encoding transcription factors were expressed in the endometrial epithelial cells while several exhibited differential expression patterns between control and LPS samples (Figure 3A). Transcription factors with intriguing mRNA over-expression levels include *AHR*, *BACH2*, *EGR2*, *EGR3*, *FOXA1*, *FOXO1*, *NR4A2*, and *STAT1*, while *MYB* was under-expressed. We examined the DNA methylation status of regions containing motifs for these transcription factors at a genome-wide level, inferred from our previous study [38]. A gradual demethylation at transcription factor-bound regions that commenced after LPS stimulation suggests a striking correlation between timing of demethylation and onset of gene expression of genes encoding transcription factors (Figure 3B,C).

## 3. Discussion

Initiation of transcription is the most important level of controlling gene expression in eukaryotic cells. Here, an in silico annotation was performed of the transcription factor binding site (TFBS) consensus sequence motifs that were present in differentially methylated regions (DMRs) resulting from LPS-treatment of bovine endometrial epithelial cells (bEECs), an in vitro model for endometritis. To achieve this, an enrichment analysis was first performed followed by a thorough in silico annotation of the identified motifs using the Jaspar database of transcription factor binding profiles [32]. In the 1291 DMRs that we identified by Reduced Representation Bisulfite Sequencing (RRBS) [38], we found that 145 TFBSs consensus sequence motifs were enriched. Based on their presence in hyper-methylated and hypo-methylated DMRs, the binding motifs were classified and then the enriched sequence motifs were further annotated regarding their regulatory nature. The majority were located in promoters, followed by enhancers. We next scrutinized the TFBSs consensus sequence motifs that were enriched in promoters of significant differentially expressed genes (DEGs) identified by RNAseq of the same cells. This analysis led to the identification of 27 such promoters [34]. Next, we defined co-localization between the five most significantly enriched transcription factors with most hits (>20% of the total DMRs) and other TFBS consensus sequence motifs. In hyper-methylated regions, binding motifs of transcription factors: ARNT::HIF-1α, PAX2, NRF1, HLTF, and MEIS1 were colocalized. In contrast, AHR:: ARNT, ARNT::HIF-1α, NRF1, TCFL5, and HES1 motifs were colocalized in hypo-methylated regions. It is notable that these transcription factors have been functionally validated as regulators of transcription in inflammatory and hypoxia-related pathways. The activity exerted by basic helix-loop-helix (BHLH) transcription factors can alter the epigenotype to promote specific cell physiology [44,45]. The enrichment of detected transcription factor motifs observed here supports the notions that they are either directly involved following LPS-associated DNA methylation changes or that their binding is influenced by differential methylation and hence relevant for gene expression changes. The validation of such direct functional effects remains to be determined.

We further analyzed the pattern of genome-wide methylation and their relative mean expression level of the top TFBSs. All these top 10 TFBSs were significantly hypo-methylated and expression of two of the genes, *HIF-1A* and *STAT1*, respectively, responded to LPS-treatment by increased mRNA levels. The signal transducer and activator of transcription (STAT) protein family are activated by Janus kinase (JAK-STAT)-dependent signaling [46,47,48]. Both STAT3 and STAT5 take part in the regulation of uterine physiology, early pregnancy [49], and the implantation process [50,51,52]. The STAT3 pathway orchestrates the inflammatory response through crosstalk with Toll-Like receptors (TLRs), inducing the production of pro-inflammatory signaling cytokines, for instance interleukin (IL)-6 [53,54]. Several family members, like STAT1, are known to be involved in LPS-activated embryo implantation and fertility processes [55,56,57], and besides JAK-dependence, these transcription factors are activated also by Leukemia inhibitory factor (LIF)-STAT signaling [58]. In our analyses, significant enrichment of STAT1, STAT2 and STAT3 consensus *cis*-acting sequence motifs were found in hyper-methylated DMRs. Stat3 and Stat5 proteins have significant biological functions during early pregnancy [49]. As mentioned above, expression of *STAT1* mRNA in response to LPS-treatment in our in vitro model was strongly increased, supporting a functional role for this transcription factor also during the early stages of endometrial inflammation. Importantly, activated *LIF* transcription was also observed after LPS-treatment of bEECs [34].

IFN regulatory factors (IRFs) belong to a family of transcription factors recognized for their roles in inflammatory responses. Both IRF-1 and IRF-3 have been shown to be stimulated by LPS, which in turn activate IFN-inducible genes [59,60]. Similarly, STAT1, another member of STAT family, plays indispensable role in the regulation of cell proliferation, development, innate immunity, and inflammatory responses [40,61].

Currently it is evident that altered gene expression patterns under hypoxic conditions during inflammation are mainly controlled at the transcriptional level. Members of the hypoxia-inducible transcription factor family include Hypoxia-Inducible Factor 1-Alpha (HIF-1α), which is a central transcription factor responding to hypoxia. Interestingly, the level of *HIF-1A* mRNA was strongly increased after LPS treatment and furthermore HIF-1α/ARNT consensus binding motifs were highly enriched in the hypo-methylated DMRs supporting a functional role for these transcription factors in this endometritis model.

A well-known functional connection between inflammation, such as endometritis, and hypoxia exists, which is largely regulated by HIF-1α [62]. Our finding that both classical hypoxia-inducible and inflammatory transcription factors appear critically activated following LPS-treatment and the enrichment of differentially methylated regulatory regions containing consensus sequence binding motifs for these types of transcription factors support a link between hypoxia and inflammation in our bovine endometritis model. Additional transcription factors that regulate gene expression under normal and hypoxic conditions in bEECs are likely to be involved and their identification and functional validation may lead to improved clinical applications to treat endometritis as well as other inflammatory diseases. Our in vitro model for bovine endometritis has allowed the identification of transcription factors binding to enriched *cis*-acting regulatory motifs following activation of inflammatory pathways like the JAK-STAT pathway, a finding that could be relevant for the understanding of functional changes that develop during development of endometritis.

These in vitro results and future in vivo studies of endometritis in cattle may also serve as a comparative model for endometritis in women.

In summary, our in silico annotation of regulatory binding motifs for transcription factors in this in vitro model for bovine endometritis will guide further functional studies aimed at characterizing the critical factors involved during bacterially induced endometritis in cattle and other mammals including humans.

## 4. Materials and Methods

### 4.1. Purification of Bovine Endometrial Epithelial Cells

Uterine horns from cows were collected at the abattoir (Lövsta SLU, Uppsala, Sweden). Only uteri from diestrus cows without visible signs of pathology were used. Within 1 h after slaughter, the endometrium was dissected and tissue pieces were incubated for 2 h at 39 °C with collagenase IV (C5138, Sigma, Saint Louis, MO, USA) and hyaluronidase (250 U/mL) (H3506, Sigma) in PBS containing 2% BSA (Sigma). To remove mucus and undigested tissue, the suspension was then filtered through 250 μm gauze and then passed through a 40-μm nylon screen that allowed fibroblasts and blood cells to pass through while retaining epithelial cells. Epithelial cells were collected and centrifuged at 170× *g* for 6 min. Cell pellets were dispersed into a single cell suspension and cultured in F-12 medium (Dulbecco’s modified Eagle’s medium, D6434, Sigma) containing 10% fetal bovine serum (FBS), 1% penicillin/streptomycin (5000 units/mL penicillin/streptomycin, Gibco, Carlsbad, CA, USA), 2 mM L-glutamine, 0. 5% liquid medium supplement (ITS), and gentamicin (5 μg/mL) and nystatin (100 U/mL). Cells were seeded in a 25 cm^2^ flask and maintained in a 5% CO_2_ incubator at 39 °C. The medium was changed every 2 days. Subcultures were performed when the epithelial cells reached 80% to 90% confluence. The purity of the bovine endometrial epithelial cell (bEEC) cultures was confirmed by flow cytometry after labeling with anti-Cytokeratin 18 antibody (cat ab 668; Abcam, Cambridge, UK) and anti-Vimentin V9 antibody (cat ab175473; Abcam). From passage 2 onwards, the purity of the epithelial cells in culture is >98% [34].

### 4.2. Methylation and Gene Expression Data

Methylation data of bovine endometrial epithelial cells (bEECs) challenged with *E. coli* lipopolysaccharide (LPS) were obtained from our previous study [38] using Reduced Representation Bisulfite Sequencing (RRBS). Specifically, significant differentially methylated regions (DMRs) were used for the identification of transcription factor binding sites (TFBSs). Gene expression data produced earlier using RNAseq from the same bEECs [34] was used to analyze expression of genes encoding transcription factors.

### 4.3. Transcription Factor Binding Sites Analyses

TFBS enrichment analysis in DMRs was performed using sequence-based Single Site Analysis (SSA) tool available in oPOSSUM-3 [42]. Since our interest was to identify TFBSs enriched in either hyper-/hypo-methylated regions, therefore, we separately provided the sequences for significant hypo- and hyper-methylated DMRs to oPOSSUM-3 along with the background/control sequences and compared to the profile of known transcription factors obtained from JASPAR database [32]. We used a non-redundant profile of all transcription factors for vertebrates only. The program was run on default parameters with 85% matrix similarity threshold. Transcription factors with Z-score > 5 and DMR hits > 4 were considered significantly enriched and used for further analysis. For control, sequences of non-significant DMRs were used. Similarly, identification of over-represented TFBSs was performed in the sequences located 1 Kbp upstream of the transcription start sites (promoter region) of significantly differentially expressed genes (DEGs) identified in our previous study [34]. The promoter sequences of all expressed genes (normalized read count > 5) were used as background. All types of genomic sequences were retrieved with R packages BSgenome.Btaurus.UCSC.bosTau8 and Biostrings.

### 4.4. Colocalized Motifs in DMRs

R packages TFBSTools [63] and JASPAR2018 was used to detect colocalized TFBSs in the DMR sequences containing match hits for significantly enriched TFBSs. DMR sequences were scanned for known transcription factor patterns represented in the form of position weight matrix (PWM) using searchSeq function in TFBSTools with a stringent score (min.score = 0.95). Colocalized motifs detected in the DMRs with methylation level <25% were filtered out from the analysis.

### 4.5. Analysis of Global Methylation Levels of TFBSs and Expression of Transcription Factors

In order to obtain global methylation levels of TFBSs for the transcription factors (AHR::ARNT, ARNT::HIF-*1α*, BACH2, EGR2, EGR3, FOXA1, FOXO1, HEY1, HIF-*1α*, MYB, NR4A2, STAT1, STAT1::STAT2), total methylated regions (100 bp) were scanned for binding sites of these transcription factors using TFBSTools with min.score = 0.85. Relative FPKM expression of transcription-factor-coding genes was deduced using SUZ12, a suitable endogenous model gene for bovine endometrium model [64,65]. Additional analysis was also performed with other endogenous genes such as TBP and ACTB. Wilcoxon signed-rank test was applied to test for differences in distributions of methylation levels between control and LPS samples.

## 5. Conclusions

Lipopolysaccharide induces differential DNA methylation patterns in *cis*-acting regulatory regions in the genome of bovine endometrial epithelial cells. Novel binding sites for transcription factors involved in inflammatory and hypoxic pathways were identified in the DMRs. We further showed co-localization of certain enriched *cis*-acting regulatory motifs and performed functional annotation of those as well as the DMRs. The regulatory impacts of these transcription factors on the long-term effects on inflammation and of endometrial receptivity and implantation will be the focus of further in vivo investigations.

In addition, we could show intriguing interplay between increased mRNA levels in LPS-treated bEECs of the mRNAs encoding the key transcription factors whose binding sites were enriched in the DMRs. Our results demonstrate an extraordinary *cis*-regulatory complexity in these DMRs having binding sites for both inflammatory and hypoxia-dependent transcription factors. Obtained data using this in vitro model for bacterial-induced endometrial inflammation have provided valuable information relevant for clinical endometritis in both cattle and humans.

## Figures and Tables

**Figure 1 ijms-25-09832-f001:**
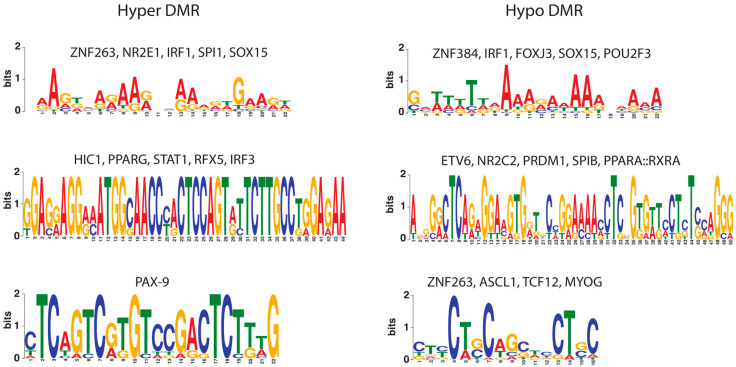
Motif enrichment in significant differentially methylated regions (DMRs). Enriched motifs found using MEME software v5.3.0 in hyper (**left**) and hypo (**right**) DMR sequences. Only the most significant motifs for known transcription factors with *p*-value < 0.01 are shown.

**Figure 2 ijms-25-09832-f002:**
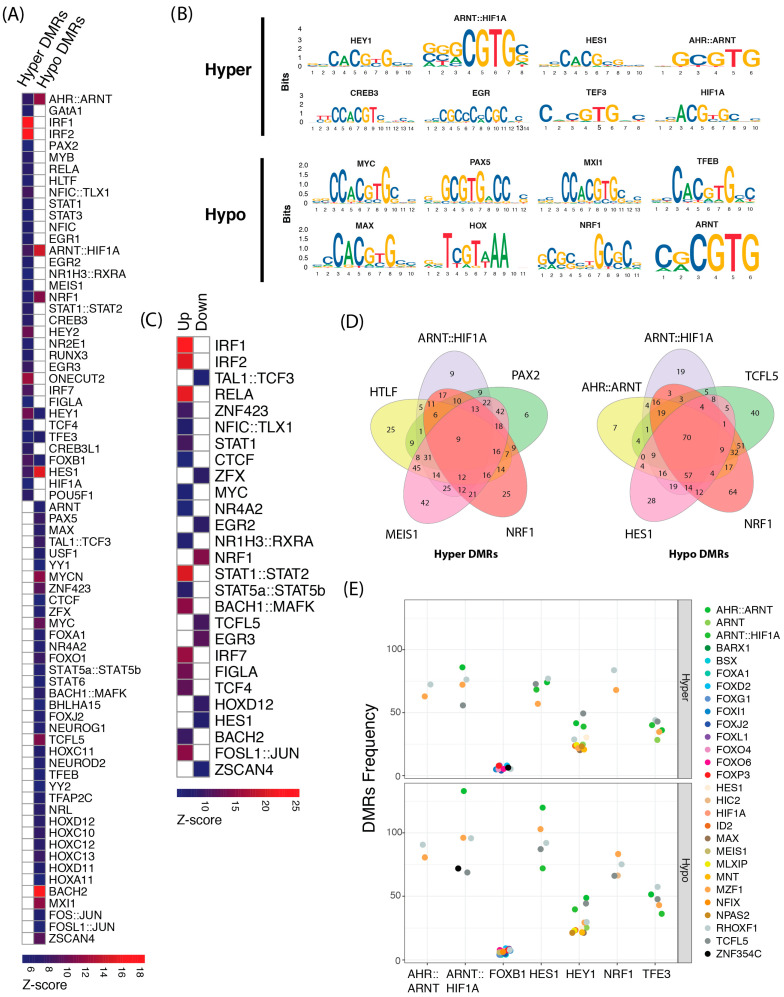
Enrichment of transcription factor binding sites in DMRs and their effect on gene expression. (**A**) Heatmap showing binding motifs of transcription factors enriched in hyper and hypo differentially methylated regions (DMRs). (**B**) Sequence logos depicting the consensus sequences of the binding-site regions in DMRs that contain CpG sites. (**C**) Heatmap showing motifs enriched in the promoter regions of significant differentially expressed genes in RNAseq data. (**D**) Hyper and hypo DMRs sharing significantly enriched transcription factor motifs. (**E**) Co-localization of flanking motifs in the DMRs with motifs enriched in both hyper and hypo as shown in heatmap (**A**).

**Figure 3 ijms-25-09832-f003:**
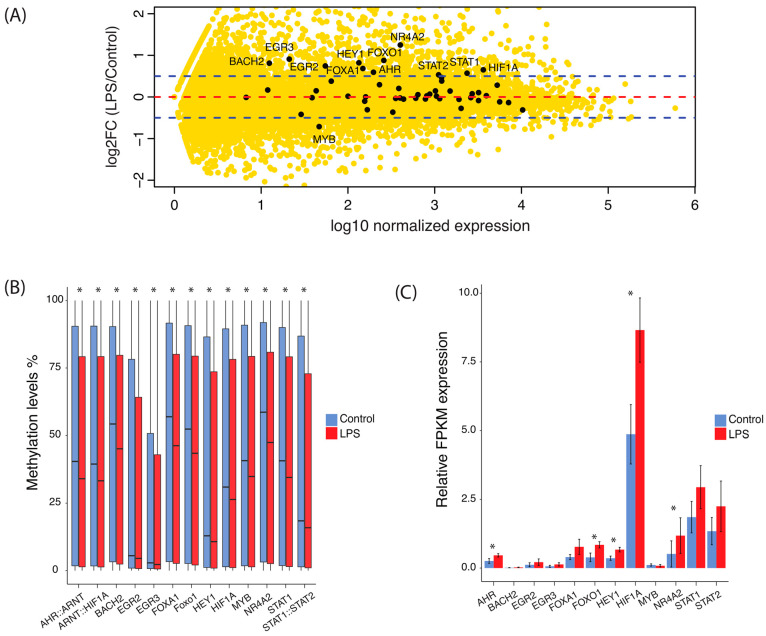
Effects of methylation changes on the activity of transcription factors. (**A**) MA-plot showing expression and fold change differences of genes between control and LPS samples. (**B**) Genome-wide methylation levels of binding sites for transcription factors with |log2 Fold change| > 0.5 as depicted in the MA-plot. Median values are represented by horizontal lines in each box. Asterisk represents significant difference between the distribution of methylation levels in control and LPS groups (Wilcoxon signed-rank test, *p* < 0.001). (**C**) Relative FPKM expression of transcription factors in control and LPS samples. Relative expression of genes shown was calculated using *SUZ12* endogenous gene from the bEECs (*p* < 0.05). Analysis with *TBP* and *ACTB* presented similar expression differences between control and LPS groups.

**Table 1 ijms-25-09832-t001:** Classification of enriched transcription factor motifs.

Class	Transcription Factors
Basic helix-loop-helix factors (bHLH)	ARNT, Bhlha15, HES1, AHR, HEY1, HEY2, MYC, MXI1, MYCN, TFE3, MAX, NEUROD2, HIF-1α, USF1, TCFL5, TAL1, TCF3, NEUROG1, TFEB, TCF4, FIGLA
Homeo-domain factors	POU5F1, MEIS1, HOXC10, HOXC11, HOXC12, HOXC13, ONECUT2, HOXA11, HOXD11, HOXD12, TLX1
Basic leucine zipper factors (bZIP)	CREB3, CREB3L1, MAFK, BACH1, BACH2, FOS, NRL, FOSL1, JUN, NRF1
C2H2 zinc finger	ZSCAN4, YY1, YY2, ZNF423, CTCF, ZFX, EGR1, EGR2, EGR3
STAT domain factors	STAT1, STAT2, STAT3, STAT5A, STAT5B, STAT6
Tryptophan clusters	IRF1, IRF2, IRF7, MYB, HLTF
Nuclear receptors with C4 zinc fingers	NR1H3, RXRA, NR4A2, NR2E1
Fork head/Winged helix	FOXB1, FOXA1, FOXJ2, FOXO1
Paired box	PAX2, PAX5
Basic helix-span-helix factors (bHSH)	TFAP2C
Runt domain	RUNX3
C4 zinc finger-type factors	GATA1
Rel homology region (RHR) factors	RELA
SMAD/NF-1 DNA-binding domain factors	NFIC

## Data Availability

The analyses were performed on RRBS raw sequencing data deposited in the European Nucleotide Archive (ENA) under accession number PRJEB36023 and on Raw RNAseq data also available in the ENA database under accession number PRJEB34011. All reference data including reference genome sequence (accession ID: GCA_000003055.4), CpG islands annotations for bosTau8 are available in the UCSC genome browser database (https://genome.ucsc.edu/). Gene annotations (Accession ID: GCA_000003055.3) are available from the Ensembl genome database (https://www.ensembl.org/).

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
