# Peer review of "Enrichment of Cis-Acting Regulatory Elements in Differentially Methylated Regions Following Lipopolysaccharide Treatment of Bovine Endometrial Epithelial Cells"

_ijms, 2024, doi:10.3390/ijms25189832_

Round 1
Reviewer 1 Report
Comments and Suggestions for Authors
This study describes the enriched differentially methylated regions and associated gene transcription in bovine endometrial epithelial cells after LPS stimulation. It provides important information to explain how endometrial cells respond to E.coli-LPS stimulation which simulates pathogen infection in endometrium in gene transcription levels. However, this reviewer suggests that the current manuscript lacks critical information about the endometrium and endometriosis in sections of introduction, discussion and methods & materials, which shall be revised accordingly.
Major comments
1. Introduction: The main aim of this study is to clarify the endometrial inflammatory responses, but the introduction section provides very few background knowledge to introduce endometrium and endometriosis. Endometriosis is involved in the interaction among several types of endometrial cells: endometrial stromal cells, luminal epithelial and glandular epithelial cells, dendritic cells, etc under pathogen infection. The cellular response of endometrial epithelial cells to exposure of pathogens is one factor of the whole network of endometrial inflammatory responses. The authors shall explain why the endometrial epithelial cell is chosen for this study and the role of its cellular responses in endometriosis. Moreover, the LPS treatment may only simulate an acute stimulation at the early stage of infection and is not a suitable scenario to represent the whole event of endometriosis. It is important to explain these points clearly the give the exact reasonable for this study.
2. Discussion: Line 275, please explain why LPS-treatment changes STATs and IRFs regulation? What is the physiological meanings of these regulations? It is important to clarify the potent effects of LPS on endometrial epithelial cells at the acute exposure of pathogens, particularly to influence the reproductive functions of endometrial epithelial cells.
4. Discussion: Although the cellular responses of endometrial epithelial cells under LPS stimulation can not represent the whole event of endometriosis, the main findings of this study still need to discuss the role of LPS-stimulated responses in bEE cells and how they work with other endometrial cells during the event of endometriosis.
5. Methods: The cell culture methodology shall be described carefully in the methods and material section, particularly the cell culture and treatment conditions, including incubation time, culture medium, dosage of LPS, and the status of bEE cells. It is also important to state the cyclic status and reproductive conditions of cows that denoted the bEE cells.
Minor comments
1. Some editing mistakes, such as the deletion line in Line 96, shall be removed. The website presented in Line 101 is not necessary. HIF-1α is presented as a wrong symbol for α. Please check and correct these editing mistakes thoroughly.
2. The resolution of Figure 1/3 is of poor quality; please improve it.
Comments on the Quality of English LanguageNA
Author Response
Reviewer 1: IJMS-3197815.
Comments and Suggestions for Authors
This study describes the enriched differentially methylated regions and associated gene transcription in bovine endometrial epithelial cells after LPS stimulation. It provides important information to explain how endometrial cells respond to E.coli-LPS stimulation which simulates pathogen infection in endometrium in gene transcription levels. However, this reviewer suggests that the current manuscript lacks critical information about the endometrium and endometriosis in sections of introduction, discussion and methods & materials, which shall be revised accordingly.
Response: We thank this reviewer for constructive criticism and good suggestions. We have included new sections in introduction, discussion and materials and methods as well as explanatory statements about the biology of the endometrium and endometritis which is the inflammatory disease in question. See page 5-7, lines 119-153
Major comments
Comment 1. Introduction: The main aim of this study is to clarify the endometrial inflammatory responses, but the introduction section provides very few background knowledge to introduce endometrium and endometriosis. Endometriosis is involved in the interaction among several types of endometrial cells: endometrial stromal cells, luminal epithelial and glandular epithelial cells, dendritic cells, etc under pathogen infection. The cellular response of endometrial epithelial cells to exposure of pathogens is one factor of the whole network of endometrial inflammatory responses. The authors shall explain why the endometrial epithelial cell is chosen for this study and the role of its cellular responses in endometriosis. Moreover, the LPS treatment may only simulate an acute stimulation at the early stage of infection and is not a suitable scenario to represent the whole event of endometriosis. It is important to explain these points clearly the give the exact reasonable for this study.
Response 1: We thank the reviewer for this suggestion and have added information about the endometrium and about endometritis in the introduction and further explained the endometrial epithelial in vitro model for endometritis. Note that the two types of epithelial cells in the bovine endometrium are present at different levels where between 80-90% are glandular epithelial cells and 10-20% are luminal epithelial cells. We have added a statement about this in both the introduction and materials and methods and a new reference to Pereira et al (ref 37). During the purification, fibroblasts and stromal cells are removed. See pages 5-6, lines 124-146
We agree with the reviewer that our in vitro model only resemble the initial acute phase of the disease and have added statements regarding this in the introduction and in the discussion sections. See page 6, lines 139-141; page 13, line 297
Comment 2. Discussion: Line 275, please explain why LPS-treatment changes STATs and IRFs regulation? What is the physiological meanings of these regulations? It is important to clarify the potent effects of LPS on endometrial epithelial cells at the acute exposure of pathogens, particularly to influence the reproductive functions of endometrial epithelial cells.
Response 2: Thank you for asking us to clarify these important issues. Both these families of transcription factors are known to be activated in response to inflammation in many different cell types including epithelia. The physiological meaning of this is that STATs and IRFs regulate transcription of a large set of genes during inflammation. We believe that we have discussed their role but we have added additional comments regarding this and stressed how this will negatively influence implantation of the fetus and also influence other critical immunological pathways during pregnancy. See page 12, lines 284-303.
Comment 3. Discussion: Although the cellular responses of endometrial epithelial cells under LPS stimulation cannot represent the whole event of endometriosis, the main findings of this study still need to discuss the role of LPS-stimulated responses in bEE cells and how they work with other endometrial cells during the event of endometriosis.
Response 3: We certainly agree with the reviewer that the LPS treatment in our in vitro model does not represent the entire set of events during endometritis. The purified cells largely consist of glandular epithelial cells. We have made a clarifying statement regarding this in the introduction and commented briefly in discussion regarding the complexity of this inflammatory disease and the limitations of the in vitro model. See page 6, Lines 126-131; page 14, lines 327-330
Comment 4. Methods: The cell culture methodology shall be described carefully in the methods and material section, particularly the cell culture and treatment conditions, including incubation time, culture medium, dosage of LPS, and the status of bEE cells. It is also important to state the cyclic status and reproductive conditions of cows that denoted the bEE cells.
Response 4: Thank you for pointing this out. We have now added a thorough explanation of the purification of bEEC in the material and method section. See pages 14-15, lines 334 - 353
Minor comments
Comment 1. Some editing mistakes, such as the deletion line in Line 96, shall be removed. The website presented in Line 101 is not necessary. HIF-1α is presented as a wrong symbol for α. Please check and correct these editing mistakes thoroughly.
Response 1: We have made these corrections. Note that the alpha symbol was changed when the editorial office produced a pdf.
Comment 2. The resolution of Figure 1/3 is of poor quality; please improve it.
Response 2: The resolution of the figures was at highest possible which were submitted separately during the process. We are assuming that the low resolution in the version used during the review was result of converting images from the manuscript file (Word document). This can be corrected by the technical staff at the editorial office
Reviewer 2 Report
Comments and Suggestions for Authors
Authors have provided an excellent work from an EU funded project. The manuscript is well organised and well presented.
Authors must explain the home keeping genes, the control genes and the gene expression measurement in a more analytical way.
The mexhanism of endometritis and the genes related are alos need to be explained.
The authors must reduce similarity. Especially the first source is used at level 7%. All other sources are used at 1%. Similarity should be less than 20%. not any source should be used more than 1%.
Some recent references should be added.
Author Response
Authors have provided an excellent work from an EU funded project. The manuscript is well organised and well presented.
Response: Thank you for this comment!
Comment 1: Authors must explain the home keeping genes, the control genes and the gene expression measurement in a more analytical way.
Response 1: Thank you for this suggestion. It is important to note that all the details about the RNA-sequencing and validations are described in our earlier publication (Guo et al. PLoS ONE 14(9): e0222081 (2019), where we performed transcriptomics and analysed mRNA levels in bEECs before and after LPS treatment. A set of house-keeping genes was then identified and described. See references to Ref34: Lines 143, 149, 162, 216, 241, 268, 298, 353, 360 and 375
Comment 2: The mexhanism of endometritis and the genes related are alos need to be explained.
Response 2: Thank you for this suggestion. We have added a more indepth information about the potential mechanisms underlying endometritis as well as the genes involved in this inflammatory disease. See pages 5-7, lines 119-153.
Comment 3: The authors must reduce similarity. Especially the first source is used at level 7%. All other sources are used at 1%. Similarity should be less than 20%. not any source should be used more than 1%.
Response 3: Thank you for pointing this out. We have acted on this a reduced the similarities as requested. See Pages 3-4, lines 75-77; 87-88; 94-98; 108-110; 113
Comment 4: Some recent references should be added.
Response 4: We have added references as requested. Ref 35, 37.
Reviewer 3 Report
Comments and Suggestions for Authors
1. The authors performed a series of study of bovine endometrial epithelial cells from differential gene expressions, then differential DNA methylation of genes associated with inflammation and endometrial function, and finally to cis-acting regulatory elements in DMRs specifically for access by inflammatory and hypoxia-dependent transcription factors. The transcriptomics, epigenomics and bioinformatics results provide a depth for the complexity of regulatory transcriptional machinery during inflammation infection in bovine endometrial epithelial cells.
2. The study lacks a functional validation. For example, Figure 2C showed 27 TFBSs enriched both in DMRs and in the promoters of DEGs and Figure 3C showed that in the transcription factor motifs enriched DMR, some of 37 genes encoding transcription factors were expressed differentially. In the 27 and 37 genes, some are overlapped such as BACH2, EGR2, EGR3, and STAT1. If a functional assessment such as Western blot, immunostaining, or ELISA to define the changes of the effectors downstream the overlapped transcription factors would increase the credit of the study.
Comments on the Quality of English LanguageMinor editing of English language required.
Author Response
Comment 1: The authors performed a series of study of bovine endometrial epithelial cells from differential gene expressions, then differential DNA methylation of genes associated with inflammation and endometrial function, and finally to cis-acting regulatory elements in DMRs specifically for access by inflammatory and hypoxia-dependent transcription factors. The transcriptomics, epigenomics and bioinformatics results provide a depth for the complexity of regulatory transcriptional machinery during inflammation infection in bovine endometrial epithelial cells.
Response 1: We thank the reviewer for the concise summary of our studies.
Comment 2: The study lacks a functional validation. For example, Figure 2C showed 27 TFBSs enriched both in DMRs and in the promoters of DEGs and Figure 3C showed that in the transcription factor motifs enriched DMR, some of 37 genes encoding transcription factors were expressed differentially. In the 27 and 37 genes, some are overlapped such as BACH2, EGR2, EGR3, and STAT1. If a functional assessment such as Western blot, immunostaining, or ELISA to define the changes of the effectors downstream the overlapped transcription factors would increase the credit of the study.
Response 2: We appreciate the comment for the lack of functional validation in the present study. The scope of the present study is however exclusively at the bioinformatics level. We would like to point out that the enrichment of transcription factor binding sites and in certain cases, direct effects at the mRNA levels of such transcription factors, as reported earlier in our own studies (Guo et. al. PLoS ONE 14(9): e0222081 (2019); Jhamat et. al. BMC Genomics 21:385, 2020), and highlighted in the present study, are only supporting the notion that there are direct functional effects. The set of inflammatory transcription factors that we here show enrichment for their respective biding sites are in most cases consistent with their well-known functional role during inflammation. Thus, the possible functional effects correlating differential mRNA levels with differential DNA methylation that were observed in our earlier publications (Guo et. al. PLoS ONE 14(9): e0222081 (2019); Jhamat et. al. BMC Genomics 21:385, 2020), are corroborated in the present study and strengthened by the fact that the mRNA levels were increased in comparison to the untreated control cells (Guo et. al. PLoS ONE 14(9): e0222081 (2019); Jhamat et. al. BMC Genomics 21:385, 2020; present study). We have added clarifying statements in the discussion of the revised manuscript. See e.g. pages 21-22, lines 310-311; 316-319
We agree with the reviewer that functional validation is important. However, such experiments are beyond the scope of the present study. For your information, more direct functional studies using transient transfection of bEECs with reporters with such promoters are planned and results to confirm such potential direct functional effects will be done as well as biochemical experiments. Moreover, in vivo assessment of the transcriptome and epigenome in Holstein cows clinically diagnosed to be affected by endometritis in comparison to healthy cows are under way. All these sets of experiments are time-consuming and will be reported in future follow-up manuscripts.
Comments on the Quality of English Language
Comment 3: Minor editing of English language required.
Response 3: We have allowed a native English speaking colleague check the English language for improvements.
Round 2
Reviewer 1 Report
Comments and Suggestions for Authors
This reviewer is satisfied with this revised manuscript and suggests acceptance.